# OPEN-DOMAIN VISUAL ENTITY LINKING

## ABSTRACT

We introduce the task of Open-domain Visual Entity Linking (OVEN), targeting a wide range of entities including animals, plants, buildings, locations and much more. Given an image (*e.g.*, an image of an aircraft), a text query ('`What is the model?`' or '`What is the airline?`'), and a multi-modal knowledge base (*e.g.*, Wikipedia), the goal is to link to an entity (BOEING-777 or EVA AIR) out of all entities in the knowledge base. We build a benchmark dataset (OVEN-Wiki), by repurposing 14 existing image classification, image retrieval, and visual QA datasets. We link all existing labels to Wikipedia entities when possible, using a state-of-the-art entity linking system and human annotators, creating a diverse and unified label space. OVEN is a rich and challenging task, which requires models to recognize and link visual content to both a small set of seen entities as well as a much larger set of unseen entities (*e.g.*, unseen aircraft models). OVEN also requires models to generalize to previously unseen intents that may require more fine-grained reasoning ('`What is the model of the aircraft in the back?`'). We build strong baselines based on state-of-the-art pre-trained models and find that current pre-trained models struggle to address the challenges posed by OVEN. We hope OVEN will inspire next-generation pre-training techniques and pave the way to future knowledge-intensive vision tasks.

## 1 INTRODUCTION

Recent interest in knowledge-intensive visual applications such as KVQA (Shah et al., 2019) and OK-VQA (Marino et al., 2019) has demonstrated the value of grounding images to knowledge bases such as Wikipedia. However, while models for these applications focus on integrating information from the knowledge base, there has been little focus on systematic, broad-coverage approaches to the grounding problem itself. Existing work typically combines various closed-set classifiers (*e.g.*, ImageNet (Russakovsky et al., 2015), COCO (Lin et al., 2014), *etc.*) in an ad-hoc manner, without a clear formulation of image grounding as a general task.

In this paper, we propose and formally define the task of **O**pen-domain **V**isual **En**tity Linking (OVEN), with the goal of building vision systems that ground visual content to entities in large-scale multi-modal knowledge bases (such as Wikipedia). In contrast to existing informal setups that rely on closed-set classifiers in an adhoc way, OVEN is open-domain where predictions cover a large space of entities governed by a knowledge base. The OVEN task takes as input with an image, a text query[1] that expresses intent with respect to the image, and a knowledge base which contains the entire set of entities, along with supporting text descriptions and a relevant set of images for each. Given these inputs the goal is to predict an entity that is both physically present in the input image as well as satisfies the unambiguous intent expressed in the query. For instance, given the same image of an aircraft, different text queries such as '`Which model is this?`' or '`Which airline is this?`' can lead to different answers, *i.e.*, BOEING-777 or EVA AIR. A strong OVEN model should learn to use all three inputs, *i.e.* the query, the image and the multi-modal knowledge base when making predictions. Figure 1 illustrates the input-output mapping of OVEN.

By connecting images to entities in the knowledge base, OVEN creates a universal entity space that organizes existing image labels. This setup poses challenges in the form of (1) generalization to UNSEEN *entities* and (2) generalization to UNSEEN *queries*. For example, classic object recognition systems trained on FGVCAircraft (Maji et al., 2013) can recognize the ninety-six aircraft categories

---

[1]A query can be expressed in different formats; in this paper, we choose to use a question to reflect the intent.

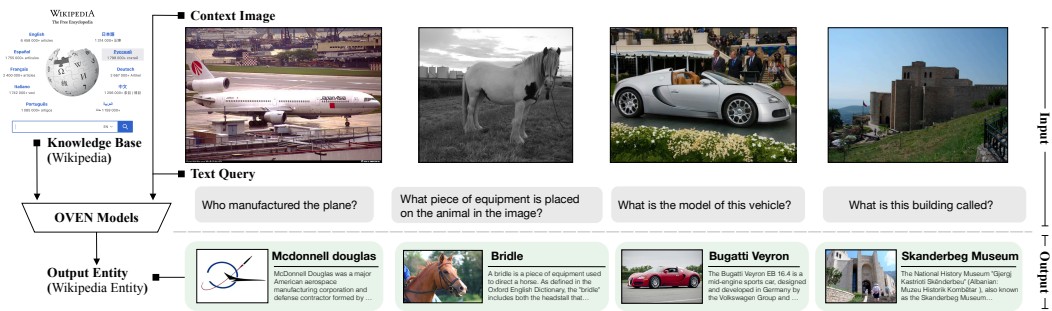

Figure 1: An illustration of the proposed OVEN task. Examples on the right are sampled from the constructed OVEN-Wiki dataset. OVEN aims at linking entities *physically* presented or revealed in the image.

defined in FGVCAircraft, but would necessarily fail to recognize BOEING-787, as it does not belong to the pre-defined categories. In contrast, models for OVEN are required to recognize entities that were UNSEEN in the training data. This requirement reflects a more realistic scenario, since there will never be enough training data to cover all knowledge base entities, especially when the number of knowledge base entities is constantly growing.[2] OVEN also evaluates a model's ability to understand UNSEEN queries, since it is impossible to observe all *text queries* to a real-world KB during training.

To benchmark OVEN, we construct OVEN-Wiki by repurposing 14 existing image classification, image retrieval, and visual QA datasets, and grounding all labels to the most prominent knowledge base, Wikipedia. The training, validation, and testing splits are formed in a way such that models need to generalize to entities that are unseen during training. Grounding labels from different datasets into Wikipedia and combining datasets is challenging because of the ambiguity of language. For example, due to language polysemy, 'Yoke' can represent a wooden beam or part of the construction of a garment. 'Tornado' can be a weather phenomena or a type of airplane (PANAVIA TORNADO). To reduce such ambiguity in the grounding, we take multiple steps to refine the labels, including the use of a state-of-the-art textual entity linking system (De Cao et al., 2020). For more accurate evaluation, a subset of examples is thoroughly annotated by human annotators: entity linking errors are corrected and the ambiguous queries are rewritten so that no other objects can be the answer. To ensure that the dataset is computationally manageable for the community, we use a 100k subset of Wikipedia entities.

Using multiple types of information (image, input query and multi-modal knowledge) is important for succeeding on OVEN, but existing pre-trained models like CLIP (Radford et al., 2021) or SimVLM (Wang et al., 2021) cannot use all of the available information natively. We experiment with baselines applying different combinations of strong pre-trained models and conduct error analysis to show possible headroom. The contributions of the paper are as follows:

- Towards the broader goal of linking visual entities to large open-domain KBs, we formalize the task of Open-domain Visual Entity Linking (OVEN)
- We construct the first large-scale visual entity linking benchmark, OVEN-Wiki, by re-purposing 14 existing datasets with all of their labels grounded to Wikipedia. A subset of the dataset is annotated by human annotators for high quality evaluation.
- We build strong baselines based on state-of-the-art pre-trained models and find that the best performing systems are the ones that better leverage multi-modal knowledge.

## 2 TASK FORMULATION FOR OVEN

The proposed task of open-domain visual entity linking relies on inputs that consist of text queries, images and a multimodal knowledge base (KB), all corresponding to a unified label space. The input image-text pairs $x = (x^p, x^t)$ include a text query $x^t$ expressing intent with respect to the corresponding image $x^p$. Given a unified label space $\mathcal{E}$ which defines the set of all possible entities, the input knowledge base $\mathcal{K} = \{(e, p(e), t(e)) | e \in \mathcal{E}\}$ is a set of triples, each containing an entity $e$, its corresponding text description $t(e)$ and a (possibly empty) set of relevant images $p(e)$. For instance, an entity $e =$ SABATIA CAMPESTRIS would have a corresponding textual description $t(e) =$ 'Sabatia campestris is a species of Sabatia ...' and a set $p(e)$ containing one

---

[2]English Wikipedia has grown from 3M to 6.5M articles in the last decade and continues to grow.

or more images from the corresponding Wikipedia page[3] of SABATIA CAMPESTRIS. We consider the combination of $t(e)$ and $p(e)$ the *multimodal knowledge* for the entity $e$.

In the current framework, we focus on linking entities that are *physically* present in the image.[4] This means that given an image, the target label is an entity that is present in the image and answers the query. There could be many different modeling choices for OVEN. As one modeling choice, we can learn a function $f_\Theta$ to maximize the score of the target entity for the given image-query pair, using multimodal knowledge from the knowledge base.

Then, given a test example $x = (x^p, x^t)$ and the knowledge base of triples $\mathcal{K}$, the function is used to make a prediction,

$$e' = \arg\max_{e \in \mathcal{E}} f_\Theta(x^p, x^t, p(e), t(e)) \tag{1}$$

where a large set of entities in $\mathcal{E} = \mathcal{E}_{\text{SEEN}} \bigcup \mathcal{E}_{\text{UNSEEN}}$ is a combination of SEEN and UNSEEN entities. Note that all examples available during the training stage have entities in $\mathcal{E}_{\text{SEEN}}$.

## 2.1 COMPARING OVEN TO OTHER TASKS

**OVEN is a challenging recognition task** By setting the text query to 'What is the main object?', we can recast OVEN into a general object recognition task. However, OVEN is far more challenging compared to existing benchmarks because knowledge bases such as Wikipedia can provide a unified label space. The number of entities in Wikipedia is orders of magnitude larger than conventional benchmarks. We demonstrate the challenge of using a large label space as follows—we start by selecting an image from a standard benchmark and using CLIP (without fine-tuning) to encode it, then we scan 100K Wikipedia titles and make a prediction. Figure 2a[5] shows that extending the original label space to 100K Wikipedia titles has introduced more closely related hard negative examples that turn many existing features into weaker signals. For example, models can no longer rely on just the color or the car body configuration to identify the ground-truth category DAEWOO NUBIRA, confusing it with other cars with similar configurations. Instead, fine-grained reasoning over visual attributes is required to make a correct classification.

In addition, the introduction of a large label space also makes generalization to previously UNSEEN entities critical, which is not emphasized in most image recognition benchmarks. This is important because there would not be enough training data to learn classifiers for every possible entity in a Web-scale KB such as Wikipedia, especially given that it grows over time.

Finally, knowledge bases such as Wikipedia cover entities of different granularities (*e.g.*, the airplane manufacturer or the aircraft model), such that different entities would be preferred according to different text queries. Given the scenario in Figure 2b, OVEN models should predict different entities given the same input image but different text queries, whereas recognition models can not handle this.

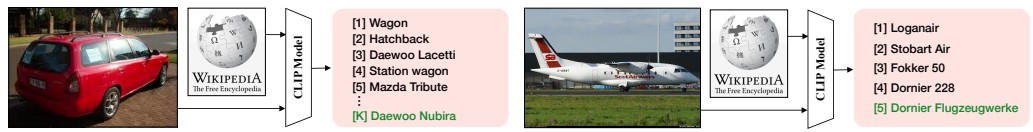

(a) Large entity sets                                    (b) Ambiguous intent

Figure 2: Why is OVEN challenging? We show that CLIP cannot solve OVEN out of the box. We ran CLIP to encode images from standard recognition datasets and retrieve Wikipedia titles from 100K Wikipedia entities. **Left: (a)** As the # of labels become large, difficult negatives start appearing. For example, DAEWOO LACETTI does not belong to the original label set. **Right: (b)** Intents can be ambiguous—DORNIER 228 is the model and DORNIER FLUGZEUGWERKE is the manufacturer.

**OVEN is a specialized Visual QA task with KB grounding** Due to the input text query, we can consider OVEN to be a specialized VQA task with KB grounding. However, OVEN is not a VQA task as OVEN models are required to output the entity name inside a KB (same as text entity linking

---

[3] https://en.wikipedia.org/wiki/File:Sabatia_campestris_Arkansas.jpg

[4] Extending this framework to entities that are not physically present in the image (e.g. the inventor of the airplane) is also valid and useful—we leave this to future work.

[5] The omitted questions of Fig. 2a and Fig. 2b (because standard CLIP model ignores it) are "what is the model of the car?" and "who manufactured this aircraft?"

tasks), while VQA models output free-form answers that can contain yes-no/counting questions. Moreover, OVEN only focuses on the entities that are present in or can be directly inferred from the image while VQA tasks can ask questions referring to entities not in the image. Another unique feature of OVEN is that it covers over 100K fine-grained entities and this encourages the models to make use of the multimodal knowledge in general KB for unseen entities.

**Knowledge Base as Input** It is important to note that the knowledge base is an input to the model. Therefore, the model should still work if the knowledge base gets updated, as the KB is a constantly evolving knowledge source. The two possible updates include: (1) editing an existing entity's text description or image set, and (2) growing the label space by adding new entities.

## 2.2 EVALUATION FOR OVEN

We evaluate OVEN with the goal of balancing performance between SEEN and UNSEEN entities using a harmonic mean, as shown below:

$$\text{HM}(\text{ACC}_{\text{SEEN}}, \text{ACC}_{\text{UNSEEN}}) = 2 \quad / \quad (\frac{1}{\text{ACC}_{\text{SEEN}}} + \frac{1}{\text{ACC}_{\text{UNSEEN}}}) \tag{2}$$

Here, $\text{ACC}_{\text{SEEN}}$ and $\text{ACC}_{\text{UNSEEN}}$ are the Macro accuracy over examples from all SEEN and UNSEEN classes, respectively. Harmonic mean equally weighs the importance of both SEEN and UNSEEN subsets, and penalizes models with a short barrel. More concrete details are provided in §3.

## 3 THE OVEN-WIKI DATASET

Based on our task formulation, we create a benchmark dataset using a comibation of automatic entity linking and human annotation. More specifically, we create OVEN-Wiki by combining 14 existing datasets originally created for image recognition, image retrieval, and visual question answering, and creating a unified label space grounded in Wikipedia entities. This dataset provides training and evaluation examples, and a Wikipedia-based KB for the proposed open-domain visual entity linking (OVEN) task. The complete list of datasets includes:

- **Image Recognition (or Retrieval) Datasets**: ImageNet21k-P (Russakovsky et al., 2015; Ridnik et al., 2021), iNaturalist2017 (Van Horn et al., 2018), Cars196 (Krause et al., 2013), SUN397 (Xiao et al., 2010), Food101 (Bossard et al., 2014), Sports100 (Gerry, 2021), Aircraft (Maji et al., 2013), Oxford Flower (Nilsback & Zisserman, 2008), Google Landmarks v2 (Weyand et al., 2020)
- **Visual QA Datasets**: VQA v2 (Goyal et al., 2017), Visual7W (Zhu et al., 2016), Visual Genome (Krishna et al., 2017), OK-VQA (Marino et al., 2019), Text-VQA (Singh et al., 2019).

These datasets generally belong to two groups: image recognition (or retrieval) which provides *diverse visual entities*, defined as the **Entity Split**; and visual QA which provides *visually-situated natural language queries*, defined as the **Query split**.

We describe data collection, filtering and curation in Appendix A.1. In the following, we present the dataset statistics, and introduce evaluation protocols.

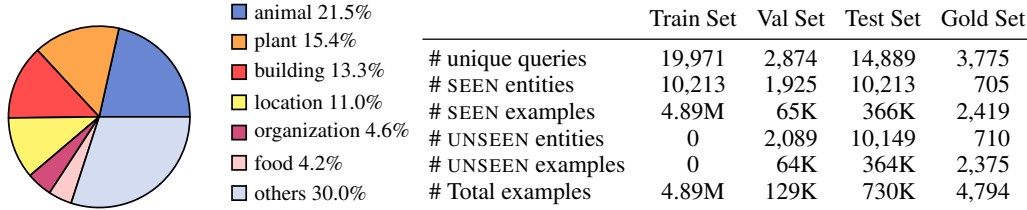

| | Train Set | Val Set | Test Set | Gold Set |
|---|---|---|---|---|
| # unique queries | 19,971 | 2,874 | 14,889 | 3,775 |
| # SEEN entities | 10,213 | 1,925 | 10,213 | 705 |
| # SEEN examples | 4.89M | 65K | 366K | 2,419 |
| # UNSEEN entities | 0 | 2,089 | 10,149 | 710 |
| # UNSEEN examples | 0 | 64K | 364K | 2,375 |
| # Total examples | 4.89M | 129K | 730K | 4,794 |

- animal 21.5%
- plant 15.4%
- building 13.3%
- location 11.0%
- organization 4.6%
- food 4.2%
- others 30.0%

Figure 3: Dataset Statistics of the OVEN-Wiki. **Left:** Distribution of super-categories of positive entities in our dataset (See Fig. 6 in Appendix for more details). **Right:** Statistics of different sets.

**Dataset Statistics** Since our goal is to evaluate generalization to both existing and novel entities, we hold out half of the entities as UNSEEN and only include them for evaluation. We also hold out examples from SEEN entities to measure SEEN performance in the evaluation. Figure 3 (left)

presents the general distribution of the super-categories for our final collection of positive Wikipedia entities. Figure 3 (right) shows detailed statistics for queries and entities for each of the training (or fine-tuning), validation, test and gold splits.[6] The number of SEEN/UNSEEN examples indicates the number of examples of which the positive entity labels are in the SEEN/UNSEEN split.

**Human Annotated Gold Evaluation** Besides evaluating on the automatically constructed dataset, we further perform a human annotation to create the gold evaluation data. Specifically, we hired over 30 dedicated annotators to validate the entity links in <image, query, answer> triplets sampled from the test split. They were asked to re-annotate the triplets, with access to the visual context, ensuring that the query leads to the correct Wikipedia entity answer. This process identified an approximate 8.04% of entity linking errors, and obtained 4,794 natural language queries, equally distributed over triplets originally sampled from the Entity and Query splits (*i.e.*, test splits). We asked the annotators to rewrite the queries so that no other object in the image can be the answer. As a result, the percentage of unique queries in the total examples (3,775 out of 4,794) as shown in Table 3 (right) is significantly higher in the gold set than the other sets, which brings more challenges to query generalization for the gold eval set. We report results using the same evaluation metrics on the gold data, with respect to SEEN and UNSEEN entities. Figure 1 provides a glance at the human annotated gold data. More details about human annotation is in Appendix A.2.

**Evaluation Protocol** To approach our mission of linking images to Wikipedia entities without imposing overwhelming computational costs, we create an *approximated* Wikipedia knowledge base. Specifically, we include a total of 85k top frequent and non-overlapping Wikipedia entities into our dataset, which sums to a total of 105k entities. This step simulates practical entity linking scenarios, where a model is required to contrast against a massive number of Wikipedia entities and infer the correct one for a query on the visual content. In this paper, all models predict among $\sim$105k candidate entities. As stronger models emerge, we can consider a larger knowledge base in the future work.

We measure the model's performance using both the Entity Split (ES) and Query Split (QS). Specifically, we first compute the harmonic mean of Macro F1 over examples from SEEN and UNSEEN classes, as $\text{Acc}_{\text{ES}}$ and $\text{Acc}_{\text{QS}}$, according to Equation 2. Then we further calculate the harmonic mean between splits $\text{HM}(\text{Acc}_{\text{ES}}, \text{Acc}_{\text{QS}})$ to reward models that do well on both splits.

We use the validation data, which contains examples from subsets of both SEEN and UNSEEN entities, for model selection, and report the overall evaluation. For final performance, we measure performance on the test split, which covers $\sim$98% positive entities (about half of them are unseen). We also report results on the Gold set, to measure model's generalization to a more diverse set of human queries.

## 4 BASELINE MODELS FOR OVEN

As performance on UNSEEN entities and text queries is important for OVEN, we focus on applying pretrained multimodal models which have previously reported promising zero-shot generalization.

### 4.1 EXISTING PRE-TRAINED MULTI-MODAL MODELS

We consider the following two representative pre-trained models.

**CLIP: Image-to-Text Dual Encoder** CLIP (Radford et al., 2021) is the most popular dual encoder model for images and text, where one encoder encodes images and the other encodes text. The biggest advantage of a dual encoder model is that the embedding of all candidates in the knowledge base can be pre-computed, so that when a new example arrives, only a dot product is required to compute the similarity between query input and entity representations. Based on the different combination of input information and entity information employed, there are fours variants of CLIP models, which make predictions based on cosine similarities between $<x^t, t(e)>$ (CLIP T2T), $<x^t, p(e)>$ (CLIP T2I), $<x^p, p(e)>$ (CLIP I2I), and $<x^p, t(e)>$ (CLIP I2T). Without explicit notation, we refer CLIP I2T model as the default CLIP models. Note that all CLIP variants are depending on partial information to pair queries with entities. However, CLIP's large-scale contrastive pre-training—with 400M image-caption pairs—makes it a strong baseline in practice.

---

[6]Note that the models do not know which entities are present in the val/test/gold set, and must scan through the whole KB to make predictions.

**SimperVLM: Image+Text Encoder with Text Decoder (SimpVLM)** SimVLM (Wang et al., 2021) is a pretrained model with an encoder that can jointly encode text and images, and outputs text with a decoder. SimVLM has demonstrated strong performance on vision and language tasks, including Visual QA. Using this model for OVEN is straight-forward, as we use the encoder to encode inputs $(x^p, x^t)$, and use the decoder to generate the target entity $e$. However, the generated text is unconstrained and might not belong to the set of possible entities $\mathcal{E}$. In addition, this model also is not capable of using the entity images $p(e)$. For this paper, we implement our version of SimVLM (Wang et al., 2021) (dubbed SimplerVLM) where we remove the Convolutional Neural Networks (*i.e.*, Transformer only) and train the simplified model on the combined datasets of Fit400M (Yu et al., 2022) and C4 (Raffel et al., 2020). The total # of image-text pairs used for training is similar to the CLIP model's pre-training dataset, and the # of text-only examples is the same as Wang et al. (2021).

Note that these two models do not make use of all information provided by OVEN. Next we extend these two pre-trained models to construct our baseline models that cover more types of information.

## 4.2 OUR BASELINE MODELS: MIXED-MODALITY DUAL ENCODERS

In the following, we propose baseline models that can use all the information provided for a given $(x, e)$ pair in OVEN: $x^p, x^t, p(e), t(e)$. Note that it is computationally infeasible to score all possible entities in $\mathcal{K}$ through a joint encoder of all components of the input and all components of each candidate entity, due to the large size of the candidate set. We therefore limit attention to scoring functions that decompose as dual encoders, *i.e.*, $f_\Theta(x^p, x^t, p(e), t(e)) = f_\theta(x^p, x^t) \cdot f_\phi(p(e), t(e))$. Given that both the input pair $(x^p, x^t)$ and the knowledge base information $(p(e), t(e))$ are mixed modality, we call these models Mixed-Modality Dual Encoders. Note that the parameters of Dual Encoder $\Theta$ have two components, *i.e.*, left encoder parameters $\theta$ and right encoder parameters $\phi$, which are not shared for all the Mixed-Modality Dual Encoders in this paper.

We study three baseline mixed-modality dual-encoders:

- **Dual SimplerVLM (D-SimpVLM)** converts SimplerVLM into a mixed-modality dual encoder by using its encoder to embed the paired image and text information into a joint space.

- **CLIP Fusion** adopts the pre-trained CLIP model as the featurizer to develop this system, via adding a 2-layer Multi-Modal Transformer on top of the CLIP image and text features.

- **CLIP2CLIP** adopts the pre-trained CLIP model via introducing a minimum set of non pre-trained parameters, only at for re-weighting the similarity scores. Particularly, it computes the cosine similarity between $<x^p, t(e)>$, $<x^t, p(e)>$, $<x^p, p(e)>$, and $<x^t, t(e)>$, using the image and text encoders of CLIP, respectively. Then it aggregates these similarities by multiplying them with a learnable vector that reflects importance weights.

For all three models, their left encoders and right encoders are using the same architecture, but differently parameterized. We fine-tune all their parameters (pre-trained or not) on the OVEN-Wiki.

**Implementation Details** We process all the images in our dataset by resizing them to $224 \times 224$, and use standardization to normalize them. For natural language text, we perform tokenization based on the adopted pre-trained model's original vocabulary (*e.g.*, CLIP). For SimplerVLM, we employ the vocabulary from T5 (Raffel et al., 2020). Additional details on input processing, model configurations, and fine-tuning hyper-parameters are provided in the Appendix B.

## 5 EXPERIMENT

We first establish the main benchmark results in §5.1. We then conduct ablation studies to analyze models, and illustrate the key challenges of OVEN-Wiki in §5.2.

### 5.1 BENCHMARK RESULTS

**Main Results** Results on the validation set are presented in Table 1, including evaluation scores on both the Entity and Query split, as well as the overall combined score. We divide the methods into two categories, (1) zero-shot transfer models (w/ pretrained models only), and (2) fine-tuned models.

|  |  | Entity Split | | Query Split | | Overall |
|---|---|---|---|---|---|---|
|  |  | SEEN | UNSEEN | SEEN | UNSEEN | HM |
| **Zero-shot Models** | | | | | | |
| SimpVLM | B | 0.0 | 0.0 | 0.1 | 0.3 | 0.3 |
| CLIP T2T | B | 0.0 | 0.0 | 0.0 | 0.4 | 0.0 |
| CLIP T2I | B | 0.1 | 0.0 | 0.8 | 0.7 | 0.2 |
| CLIP I2I | B | 7.7 | 7.6 | 1.2 | 1.9 | 2.4 |
| CLIP I2T | B | 13.5 | 13.8 | 4.2 | 6.8 | 7.5 |
| CLIP I2T | L | 17.7 | 17.3 | 4.2 | 8.6 | 8.5 |

|  |  | Entity Split | | Query Split | | Overall |
|---|---|---|---|---|---|---|
|  |  | SEEN | UNSEEN | SEEN | UNSEEN | HM |
| **Fine-tuned Models** | | | | | | |
| SimpVLM | B | 5.0 | 0.0 | 18.4 | 0.0 | 0.0 |
| D-SimpVLM | B | 35.8 | 3.7 | 15.6 | 0.3 | 1.1 |
| CLIP Fusion | B | 33.8 | 4.5 | 19.7 | 1.1 | 3.4 |
| CLIP2CLIP | B | 18.2 | 15.4 | 10.4 | 9.3 | 12.4 |
| CLIP Fusion | L | 36.7 | 7.9 | 27.3 | 1.3 | 4.3 |
| CLIP2CLIP | L | 19.8 | 15.9 | 16.0 | 12.1 | **15.5** |

Table 1: Comparison between the zero-shot and fine-tuned models on the OVEN-Wiki **validation** set.

|  |  | Entity Split | | Query Split | | Overall | Gold Eval | | |
|---|---|---|---|---|---|---|---|---|---|
|  | Model Size | SEEN | UNSEEN | SEEN | UNSEEN | HM | SEEN | UNSEEN | HM |
| CLIP I2T | Base | 13.8 | 12.2 | 5.2 | 6.5 | 8.0 | 12.1 | 14.2 | 13.1 |
| CLIP Fusion | Base | 35.7 | 4.8 | 42.7 | 1.1 | 3.4 | 14.8 | 2.5 | 4.3 |
| CLIP2CLIP | Base | 19.1 | 13.1 | 13.0 | 9.5 | **12.9** | 19.1 | 16.1 | **17.5** |
| CLIP I2T | Large | 18.0 | 16.6 | 7.2 | 8.9 | 10.9 | 14.3 | 17.7 | 15.8 |
| CLIP Fusion | Large | 39.2 | 8.2 | 41.7 | 2.0 | 6.0 | 23.0 | 5.6 | 9.0 |
| CLIP2CLIP | Large | 20.5 | 15.8 | 13.3 | 12.9 | **15.1** | 21.0 | 18.7 | **19.8** |

Table 2: Results of top methods on the OVEN-Wiki **test** set and **gold evaluation** set.

There are several interesting observations in Table 1. First, *not all fine-tuned models are better than their pre-trained variants*, particularly on UNSEEN entities. This showcases the unique generalization challenge in OVEN-Wiki, where models cannot be over-optimized for a specific set of entities that have more training data. Next, comparing encoder-decoder models such as SimplerVLM to other methods, we observe that the former struggles to link entities accurately. We find that the zero-shot SimplerVLM model often does not produce entity names, as the pretrained model never observed Wikipeida titles. The fine-tuned SimplerVLM models do not handle unseen entities well, since the models never see the UNSEEN Wikipedia titles from the KB while training on OVEN-Wiki data.

Comparing all CLIP-based models, we observe that CLIP Fusion, where two new layers have been added on top of pretrained CLIP, shows very strong results on SEEN entities, for both the Entity and the Query splits but weak results on UNSEEN entities, thus leading to lower overall performance. The CLIP2CLIP model, on the other hand, is capable of retaining the cross-entity generalization performance while improving its prediction accuracies on SEEN entities. This suggests that adding more new (but non-pre-trained) parameters can harm the pre-trained models. Unsurprisingly, methods with large pre-trained models outperforms their smaller counterparts.

We select the best performing methods and evaluate them on the test data, as well as the human annotated gold evaluation set. Table 2 shows that the results on the Test set are generally aligned with observations on the validation set. In the gold set, annotators provide more varieties of the input queries, which can challenge the model more. Interestingly, results on the gold set suggest that methods perform better on the SEEN entities of the test split do not necessarily transfer to gold data split (*e.g.*, CLIP Fusion vs. CLIP2CLIP), perhaps due to the aforementioned variation in queries.

**Human Performance** We conduct a study to estimate the human performance on OVEN-Wiki, via requesting 3 dedicated human annotators to answer 100 examples (sampled from gold evaluation set, answers are non-overlapping). Because humans are ineffective in searching[7], we allow the annotators to use search engines (*e.g.*, Google Image Search, Wikipedia Search, etc.), as long as the annotators can provide a valid Wikipedia entity name as answer (among 105K candidates). As a result, human achieves 76% average accuracy, which is significantly higher than the best comparing systems[8].

---

[7]Even with search engines, each annotator has used 254 seconds to complete one example.

[8]For comparison, the CLIP2CLIP-Large model is evaluated on this subset, and it achieves 25% average accuracy.

| Query Repr. | | Entity Split | | Query Split | | Overall | Entity Repr. | | Entity Split | | Query Split | | Overall |
|---|---|---|---|---|---|---|---|---|---|---|---|---|---|
| $x^p$ | $x^t$ | SEEN | UNSEEN | SEEN | UNSEEN | HM | $p(e)$ | $t(e)$ | SEEN | UNSEEN | SEEN | UNSEEN | HM |
| ● | - | 18.0 | 15.0 | 8.2 | 8.4 | 11.0 | ● | - | 4.0 | 3.2 | 2.5 | 1.0 | 3.0 |
| - | ● | 0.1 | 0.0 | 2.6 | 0.4 | 0.0 | - | ● | 17.3 | 14.8 | 8.6 | 8.9 | 11.3 |
| ● | ● | 18.2 | 15.4 | 10.4 | 9.3 | **12.4** | ● | ● | 18.2 | 15.4 | 10.4 | 9.3 | **12.4** |

Table 3: Ablation studies on CLIP2CLIP-`Base` on the validation set. **Left:** Results of ablating input images ($x^p$) and input queries ($x^t$). **Right:** Results of ablating entity information ($p(e)$: entity image; $t(e)$: entity text).

## 5.2 ABLATION STUDIES AND ANALYSIS

We perform empirical studies to analyze OVEN-Wiki using the best performing model, CLIP2CLIP. Particularly, these studies are related to (1) the importance of input representations for OVEN, (2) the effects of different knowledge information for entities, (3) how fine-tuning influences generalization, (4) how the size of the Wikipedia knowledge base influences model performance, and (5) error analysis on the best performing model.

**Which input information is important for OVEN?** Table 3 (left) shows scores for CLIP2CLIP variants using different combinations of input query representations. We observe that the input image is most essential for the model when making predictions. Meanwhile, we see that query in combination with the image, can provide improvements to results on both the Entity split and the Query split, with a larger margin on the latter. Additionally, models that only look at the question can also make a reasonable guess on the Query split, but not the Entity split.

**Which entity knowledge is important for OVEN?** Table 3 (right) provides a study on different combinations of knowledge from Wikipedia, used as the representation for an entity (for the CLIP2CLIP model). Particularly, we observe that the Wikipedia entity image itself does not provide sufficient information to support strong OVEN performance, but it contains complementary information that helps the CLIP2CLIP model to better understand the entity text. As a result the best performing model is achieved via combining Wikipedia entity image and entity title, which is also serves as the de facto setting for all Mixed-Modality Dual Encoders in this paper. We note that there is other useful content on Wikipedia pages, such as the summary text. However, current experiments show that including summary text does not further improve Mixed-Modality Dual Encoders (*i.e.*, overall results drops ∼0.5%), potentially because they are often longer pieces of text with complex structure, which is out-of-domain for the pre-trained models. Appendix C included the complete study.

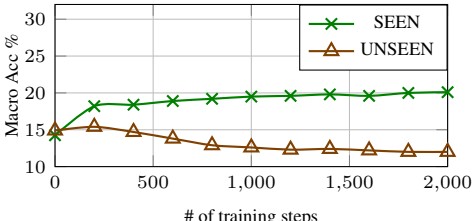

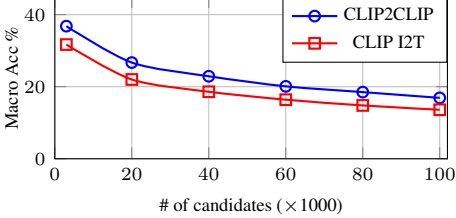

(a) Impact of fine-tuning steps for unseen entities

(b) Impact of # candidates in the KB

Figure 4: **Left: (a)** Fine-tuning CLIP2CLIP for large # of steps increases the SEEN entity accuracy but hurts the UNSEEN entity accuracy. **Right: (b)** Increasing knowledge base size makes the tasks difficult.

**Does fine-tuning always help generalization?** Figure 4 (a) shows the validation scores of the CLIP2CLIP model, during fine-tuning on OVEN-Wiki's training split. Note that we only plot the results on Entity split as the representative, because the trends are similar on the Query split. It shows that longer training schedule does not converge to better generalization performance, particularly when evaluated on the UNSEEN entities. Because of this, we employ the early stopping strategy for model selection, and pick the model with the best UNSEEN validation performance.

**How would the number of entities in KB influence the model's prediction?** Figure 4 (b) presents the accuracy of CLIP and CLIP2CLIP, as a function of the # of total candidates to retrieve from. Here, we compute the accuracy by sub-sampling the negative candidates from KB to different sizes. We observe that when the retrieval candidate entities are only the positive entities (with 0 additional negatives), the performance of both CLIP and CLIP2CLIP is significantly higher than the 100K setting. Beyond this, as the KB size increases, model accuracy decreases.

## 6 RELATED WORKS

**Learning to Recognize UNSEEN Categories** There has been a significant amount of prior work (Lampert et al., 2014; Vinyals et al., 2016; Liu et al., 2019) focusing on the generalization situation where information of novel categories are presented at test time. Zero-shot learning (ZSL) is one of such attempts that tackles learning new categories with zero images for training. To achieve such transfer, ZSL methods typically relies generating UNSEEN image classifiers based on corresponding semantic representations, in the format of manually labeled attributes (Lampert et al., 2014), unsupervised learned word vectors (Changpinyo et al., 2016), or pre-trained sentence embeddings (Kil & Chao, 2021; Radford et al., 2021). Few-shot learning (FSL) (Vinyals et al., 2016) proposes a more realistic setup, where learners have access to a limited number of visual exemplars during the model deployment. With this goal, FSL methods aim to extract the inductive bias of learning from the SEEN classes, such that the model can leverage it in learning the UNSEEN classes, to avoid severe over-fitting. Particularly, prior works either use adapted non-parametric classifiers (Snell et al., 2017; Rusu et al., 2019; Ye et al., 2020), or meta-optimized linear classifiers (Finn et al., 2017; Raghu et al., 2020) to incorporate the few-shot UNSEEN support examples. Comparing to them, our proposed task exposes different challenges as we ask the model to make the best use of open-world Web knowledge (*i.e.*, Wikipedia pages with images & figures), which contains textual semantic information and visual appearance of the entities in the open world.

**Vision and Language + Knowledge** There has been efforts in combining knowledge into vision and language tasks, such as Visual QA (Shah et al., 2019; Marino et al., 2019; Chang et al., 2022; Chen et al., 2021) and entity-focused image captioning (Liu et al., 2020; Biten et al., 2019). Among them, knowledge-based VQA is most related to OVEN, but also differ in many aspects. Specifically, Chang et al. (2022) presents a text QA dataset that requires understanding multi-modal knowledge in a KB. Shah et al. (2019) propose to perform knowledge-based question answer tasks, centered around questions that resolves relational query over public Figures. Meanwhile, Marino et al. (2019) propose to answer questions where the answer is outside of the image context, to assess model's capability in understanding real-world knowledge More recently, Chen et al. (2021) studies the zero-shot visual QA setting where some answers (out of a total of 500 frequent answers of general concepts) are unseen during the training, where a KB is supplied to assist the model in answering unseen answers. Comparing to them, OVEN steps back to the more fundamental problem of establishing the link between visual content and entity in the KB, but at a larger scale and broader coverage. We believe that stronger models developed on OVEN would benefits such knowledge intensive visual QA tasks.

**Entity Linking** Entity linking (EL) is the task of grounding entity mentions in text by linking them to entries in a given knowledge base. Supervised EL (Milne & Witten, 2008) has demonstrated its strong performance when all entities are in-distribution during the evaluation. Because KB is updating all the time, recent works (Logeswaran et al., 2019; Botha et al., 2020; Zhang et al., 2021; De Cao et al., 2020; 2022) focus on a more realistic setting where entity linking needs to be achieved in the zero-shot, with a large portion of entities (to be evaluated) completely unseen during the training. OVEN is a visual analog of zero-shot EL, and targets at developing generalizable models that recognize entities unseen in the training. Among all EL literature, visually assisted EL Zheng et al. (2022) is most relevant to this work, whose goal is to use associated image of text to improve the precision of text EL. OVEN is different as its text queries do not mention the name of the entities, which put visual understanding and reasoning into the central position.

## 7 DISCUSSION

We introduced OVEN, a task that aims to link visual content to the corresponding entities in a knowledge base. While the focus of this paper is mostly on the task and the benchmark, our results point to several interesting future avenues for representation learning research. For example, the fact that Mixed-Modality Dual Encoders outperform CLIP models suggests that future work on large scale contrastive pretraining could benefit from jointly encoding or mixing modalities. This is especially critical for unseen entities whose representations are typically learned only during pretraining. In textual entity linking, De Cao et al. (2021) has shown that it *is* possible to obtain a high-performing seq2seq model for entity linking. OVEN now provides a benchmark for evaluating such approaches for visual entity linking. In summary, we hope OVEN will drive future research on knowledge-infused multimodal representation learning via visual entity linking.

ETHICS STATEMENT

As our dataset, *i.e.*, OVEN-Wiki, is composed of existing image recognition, image retrieval, and visual question answering datasets, we have introduced minimum risk of exposing additional social bias in our data. However, OVEN-Wiki is still at the risk of inheriting existing dataset biases. As a result, we employed existing data curation strategies (Yang et al., 2020) to reduce such potential risks. Besides such risk, OVEN-Wiki also opens up new possibilities that can alleviate ethical concerns in AI systems. Specifically, OVEN-Wiki is a dataset that targets advancing research for establishing stronger grounding between the visual content and knowledge base, which can potentially contribute to building more attributed visual systems, such as a visual question answering model that produces answers based on the linked Wikipedia page, with improved interpretability and controllability.

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

## A    DATASET CONSTRUCTION, ANNOTATION, AND ADDITIONAL STATISTICS

In this section, we describes the complete details on data collection, curation, entity linking, and show additional statistics of the processed dataset (§A.1). Then we also discuss how we train annotators to annotate our task, and provide the concrete annotation interface(§A.2).

### A.1    DATA COLLECTION & PRE-PROCESSING

**Data Filtering** Some of our member datasets have been reported to include non-imageable classes, classes with undesired social bias (Yang et al., 2020), or non-entity classes (*e.g.*, numbers). Therefore, we apply a filtering process to compose our dataset, based on the individual condition of each source dataset. Overall, to create the Entity split, we first apply a general safety filter (Yang et al., 2020) to remove non-imageable labels, non-entity labels, and labels with social bias. To create the Query split, we employed three expert annotators to write heuristic policies to filter each VQA dataset, and ensure our task is focusing on entity related questions. Concretely, questions related to counting, verification, or querying non-entity attributes (*e.g.*, dates), are removed. Then we apply the same safety filter. We provide more details in Appendix.

**Entity Linking** Based on the filtered data, we developed a two-staged entity linking strategy to connect the label text to Wikipedia entities, on both Entity and Query splits. First, we obtain exact match based entity candidates by querying the Wikipedia search API (with the auto-suggestion disabled) with the raw label text. We reject candidates whose landing pages are identified as disambiguation pages. The Wikipedia API[9] automatically redirects queries (in our case, labels) matching entity aliases to their canonical form. For the labels which do not have an exact match in Wikipedia, we use a state-of-the-art text-based entity linker (*i.e.*, GENRE (De Cao et al., 2021)) to obtain top candidate Wikipedia entity names. Finally, we link the label to the top ranked entity whose landing page is not a disambiguation page.

**Preparing Multi-Modal Knowledge** Using the entity linking process described earlier, we successfully connect a total of 24,895 class labels in OVEN-Wiki to corresponding Wikipedia entities. Overall, our dataset contains 20,801 unique entities. For the Entity split data, we generate a synthetic text query based on the super-category information of the label (either provided by source dataset or mined from Wikidata[10]), using templated language. For example, iNaturalist has provided detailed supercategory annotation on each class, such as `Plantae`, `Reptilia`, *etc*. For dataset that do not provide this information, we use the super-category mined from Wikidata, which is publiclly crowd sourced and maintained. As a result, our templated query generator produces the query "`what is the species of the plant in this image?`" for the entity "`Eryngium alpinum`", whose super-category is `Plantae`. Due to space limit, we provide more explanation in Appendix. For all Wikipedia entities, we use the corresponding Wikipedia page and its associating multi-media content (*e.g.*, information box images, *etc*.) as the source of *multi-modal knowledge* about entities.

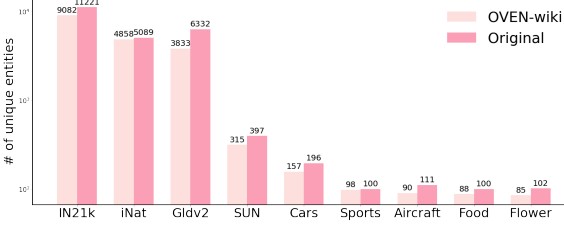

| | # Original Answers | # Entity Answers |
|---|---|---|
| VG | 50,130 | 3,460 |
| OK-VQA | 4,214 | 1,600 |
| Text-VQA | 19,500 | 3,562 |
| VQA v2 | 26,748 | 4,337 |
| Visual7W | 7,588 | 1,945 |

Figure 5: Number of unique entities on Entity split (left) and Query split (right). We compare it against the # of entities before applying pre-processing. Note that VQA datasets contain massive non-entity answers, or collapsed answers, which leads to a large reduction in numbers after pre-processing.

**Statistics on Entities** Specifically, Figure 5 shows the number of unique entities in both the Entity and Query splits, where we compare the total number of entities in each source dataset against its original population (after applied safety filter). Note that for the Google Landmarks v2 (Gldv2)

---

[9]https://www.mediawiki.org/wiki/API
[10]Available at https://www.wikidata.org/wiki/Wikidata

dataset, we employed the cleaned data split from Yokoo et al. (2020), where the total number of unique entities is significantly reduced. Because Gldv2 is automatically generated and has reported to contain noises particularly with tail entities (Yokoo et al., 2020), we removed entities with less than 50 instances for a improved precision (further reduces the # of entities in Gldv2 to ∼6k).

**Entity Super-Categories** To give more details for the Figure 3 in the main text, we further present full super-category grouping information in Figure 6. As aforementioned, we have combined entities that belongs to general groups (*e.g.*, "object", "item" groups) or unpopular groups (*e.g.*, groups with less than 5 entities) into the "others" group. We also merged some sub-categories into super-categories, *e.g.*, "location"+"park"+"lake"+"river"+"mountain"→"location", "building"+"bridge"→"building".

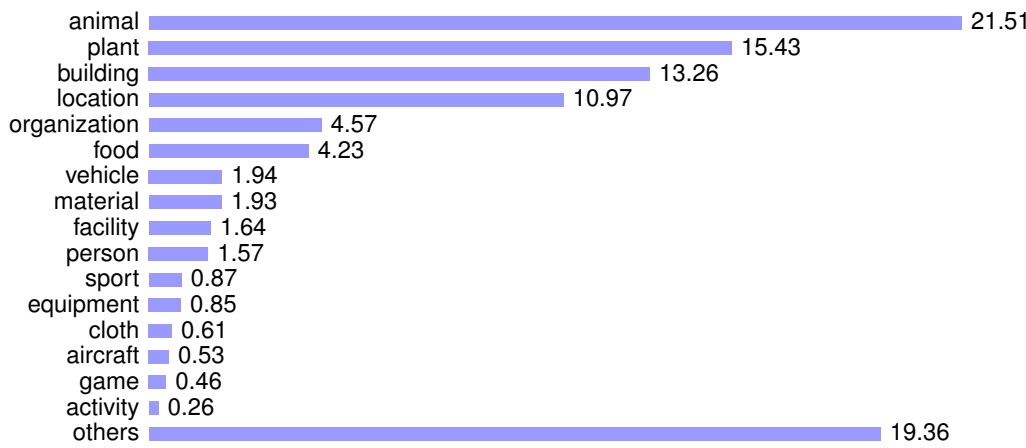

Figure 6: Distribution of the entities in our datasets (Grouped by their super category).

## A.2 HUMAN ANNOTATION PROCEDURE & INTERFACE

In order to verify the quality of OVEN-Wiki and to provide a human verified test set to evaluate on, we conduct human annotation on a subset of test set. The annotators are asked to correct the errors in the <image, query, answer> triplets. The details are as follows.

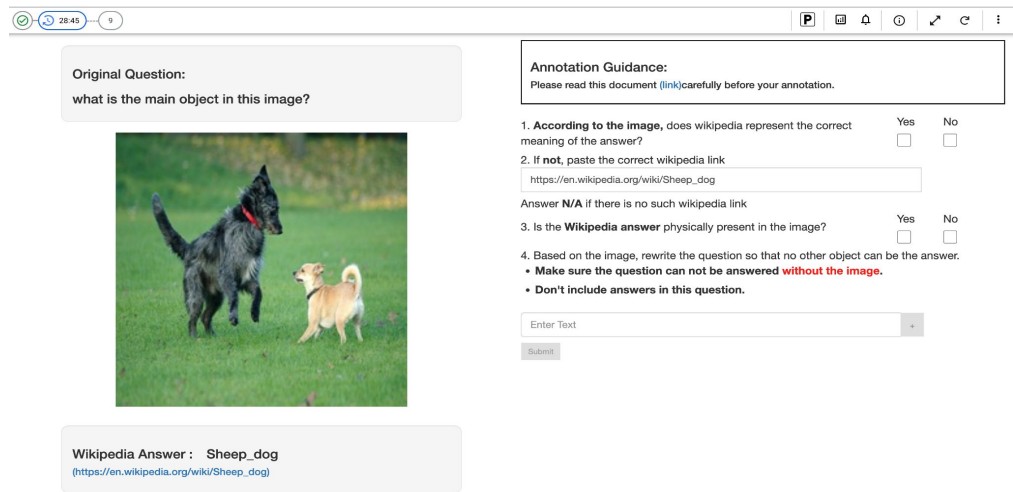

Figure 7: Annotation inferface

**Annotation interface** Figure 7 illustrates the annotation interface. The left side of Figure 7 are the input to the annotators which includes the original question, image and the answer (together with the wikipedia hyperlink). The annotators are asked to complete the following questions:

1. *Does the Wikipedia represent the correct meaning of the answer? Provide the Wikipedia link if not.*

   This question requires the annotators to correct the entity linking errors. The annotators use Google search to find the most suitable Wikipedia link if the provided one is not adequate. In our dataset, 8.4% of the entity links are reported wrong by more than 2 annotators.

2. *Is the Wikipedia answer physically present in the image.*

   This question is mainly aimed at filtering out the OCR examples which are out of our scope. One example is that the image about a wall painted with the word "love" and the linked entity is the "love" Wikipedia. In our dataset, 10.3% of the answers are reported not physically present in the image by more than 2 annotators.

3. *Rewrite the question so that no other object can be the answer.*

   The annotators will rewrite the question is the answer is wrong or ambiguous. Annotators will make sure that the question can not be answered without the image and that the answers can not be included in the rewritten questions. In our annotation, 99.9% of the questions are being rewritten.

**Instruction and Training**   We carefully design the training procedure to improve the annotation quality. We first conduct a "Self-study session" where the annotators will read the instructions and annotate a few toy examples. Then we conduct a "In-person tutorial" where we have an online video session in which we walk annotators through the full version of the instructions and discuss mistakes made in the self-study annotations. Finally we conduct a "Test exam" and the qualified annotators are accepted. In total, 30 annotators went through our training procedure and all of them were eventually accepted to work full-time on the main task.

**Quality control**   We have a three way annotations where each examples are annotated by three annotators. We were giving regular feedback on the questions the annotators may have during the annotation and pointed out mistakes identified in annotators' past answers.

On average, it took annotators 4.6 minutes to answer each question with the time consumption slightly decreasing as annotators get familiar with the task. The compensation rate for the task was set to be $17.8/hour which is higher than the minimum hourly wage in the US.

We filtered out all the examples where the wikipedia links are marked as wrong or the Wikipedia answers are marked as "Not physically present in the image".

## B   IMPLEMENTATION DETAILS OF THE BASELINE SYSTEMS

In this section, we provide complete impelmentation details on the baselines for the OVEN task.

### B.1   SIMPLERVLM & DUAL SIMPLERVLM

As aforementioned, we have simplified the architecture of original SimVLM (Wang et al., 2021) via using a Multi-Modal Transformer model (the same architecture as T5-`Base` by Raffel et al. (2020)). Specifically, we first extract non-overlapping $16 \times 16$ patches from the input image, and perform a linear transformation on them to get a sequendce of embeded patch embeddings, which is then combined with the word embeddings of a tokenized text sequence to serve as the input. The Multi-Modal Transformer then take the combined sequence and contextualize the embeddings through self-attention (Vaswani et al., 2017).

In the re-implemented SimplerVLM, we then feed the output contextualized embeddings into a Text decoder (of same architecture), which can then produce text prediction. We first pre-train this model on the combined Fit400M (Yu et al., 2022) and C4 (Raffel et al., 2020) dataset, under the PrefixLM objective defined in Wang et al. (2021). We optimize the model for 500K steps (with batch size = 4,096), using the Adafactor (Shazeer & Stern, 2018) optimizer, a initial learning rate of 0.01 and square root learning rate decay policy (with the final learning rate goes to 1e-4). Then we fine-tune the pre-trained SimplerVLM on the proposed OVEN-Wiki dataset, for 20k steps in total. The optimization is also done with the Adafactor optimizer, with the initial learning rate to be 0.001, the learning rate scheduler to be square root LR decay, and the final learning rate to be 1e-6. During

the inference, the model is then directly predicting the name of the entity, given the input text query and the context image. W count the exact matched entity name to compute the Macro F1 accuracy.

In the Dual SimplerVLM, we take the pre-trained SimplerVLM (same as above), remove its Text decoder, and use it for both the left and right encoders of the dual encoder model. Therefore, the left query encoder has the same initialization as the right entity encoder. We then fine-tune the result dual encoders model on the training set of OVEN-Wiki, under a in-batch contrastive learning objective (Radford et al., 2021), with a batch size of 4,096. Specifically, we optimize the model for 20K steps during the fine-tuning, again with Adafactor optimizer and a initial learning rate of 0.001.

### B.2    CLIP FUSION MODEL

As aforementioned, we implemented this Mixed-Modal Dual Encoder via taking pre-trained CLIP image and text encoders as featurizers. Particularly, we build two 2-layer Transformer models, on top of two CLIP models as the left and right encoder, for encoding the query representation and the entity representation, respectively. The 2-layer Transformers follows the same architecture as T5 Transformer Radford et al. (2021), but with 2 layers, 12 attention heads, with each attention head of 64 dimensions, and the embedding size of 768. We then fine-tune this composed model on the OVEN-Wiki's training data, using the same contrastive objective as Dual SimplerVLM. We optimize the model for 20K steps in the fine-tuning stage, following the same hyper-parameter configurations as Dual SimplerVLM.

### B.3    CLIP2CLIP MODEL

Different from CLIP Fusion, CLIP2CLIP is a model that adds minimum new parameters to the pre-trained CLIP encoders. Same as other models, we initialize both the query encoder and the target encoder separately with the pre-trained CLIP model. Specifically, we use the pre-trained CLIP encoders for both left and right encoders, to encode the image and text modality for both the query representation and the entity representation. We then compute the four dot product similarity scores on the <input image, target text>, <input text, target image>, <input image, target image>, and <input text, target text> pairs, which is then combined via a learnable similarity weights into one logit score. The make sure that the learnable similarity weights is initialized properly, we perform a grid search to find a roughly good similarity weights for the CLIP2CLIP model (using OVEN-Wiki's training data). Then we took this similarity weights to initialize the CLIP2CLIP model and fine-tune all parameters on OVEN-Wiki's training set, under the same contrastive learning objective. Different from other models, given that this model has most of its parameters pre-trained, we realized that it works the best to early stop the model. As a result, we only fine-tune this model for 2k steps, with an initial learning rate of 1e-4, and a square root LR decay schedule with final learning rate of 1e-6. We select the model along the way with the best UNSEEN entity performances.

## C    ADDDITIONAL EXPERIMENTS

In this section, we provide experiments that is omitted in the main text due to the space limit.

**Alternative Evaluation Metrics on OVEN-Wiki**    In addition to the primary evaluation metric, we also report the performance of dual encoder models using ranking based metrics, such as Recall@10 (R10) and Mean reciprocal rank@10 (MRR10). The results are shown in Table 4 and Table 5. Overall speaking, the results are showing the same trend as our primary metric (*i.e.*, Macro accuracy).

**Complete Studies for Entity Representation**    Table 6 (right) provides a study on different combination of knowledge from Wikipedia to use as the representation for an entity (for CLIP2CLIP model). Particularly, we observe that image is always beneficial to include as it improve the results when combined with text description. Comparing to summary, the title information can be better used by the CLIP2CLIP model. The best combination is to have both Wikipedia image and Wikipedia title as the entity representation, which is the default configuration in this paper. We note that there are many other useful content on Wikipedia pages that we have not yet exploited (*e.g.*, Section text, tables, other floating images, *etc*.), which can potentially bring poential oportunity for further improvement.

| | Model Size | Entity Split SEEN R10 | MRR10 | UNSEEN R10 | MRR10 | Query Split SEEN R10 | MRR10 | UNSEEN R10 | MRR10 | Overall HM R10 | MRR10 |
|---|---|---|---|---|---|---|---|---|---|---|---|
| **Zero-shot Models** | | | | | | | | | | | |
| CLIP T2T | Base | 0.0 | 0.0 | 0.0 | 0.0 | 6.0 | 1.4 | 3.6 | 1.1 | 0.0 | 0.0 |
| CLIP T2I | Base | 0.2 | 0.1 | 0.2 | 0.2 | 4.0 | 1.3 | 2.2 | 1.0 | 0.4 | 0.3 |
| CLIP I2I | Base | 24.9 | 12.3 | 24.7 | 12.3 | 8.5 | 3.6 | 6.0 | 2.5 | 11.1 | 5.0 |
| CLIP I2T | Base | 40.2 | 22.1 | 37.7 | 21.2 | 16.0 | 7.5 | 12.0 | 6.5 | 20.5 | 10.8 |
| CLIP I2T | Large | 46.9 | 27.0 | 44.3 | 25.9 | 15.3 | 7.2 | 13.8 | 8.5 | 22.2 | 12.2 |
| **Fine-tuned Models** | | | | | | | | | | | |
| CLIP Fusion | Base | 66.1 | 47.0 | 25.7 | 10.0 | 50.2 | 37.6 | 31.5 | 11.5 | 38.0 | 17.2 |
| CLIP2CLIP | Base | 51.3 | 29.9 | 42.7 | 24.1 | 37.6 | 20.3 | 21.9 | 12.6 | 34.9 | 19.8 |
| CLIP Fusion | Large | 69.7 | 50.6 | 37.0 | 16.0 | 58.9 | 22.5 | 45.1 | 7.3 | 49.8 | 15.4 |
| CLIP2CLIP | Large | 55.4 | 32.0 | 45.8 | 25.4 | 45.4 | 26.2 | 31.3 | 19.6 | 42.8 | 25.2 |

Table 4: Comparison between the zero-shot and fine-tuned models on the OVEN-Wiki **validation** set. Additional metrics are reported, *i.e.*, Recall@10 (R10) and Mean Reciprocal Ranks@10 (MRR10).

| | | Entity Split SEEN R10 | MRR10 | UNSEEN R10 | MRR10 | Query Split SEEN R10 | MRR10 | UNSEEN R10 | MRR10 | Overall HM R10 | MRR10 | Gold Eval SEEN R10 | MRR10 | UNSEEN R10 | MRR10 | HM R10 | MRR10 |
|---|---|---|---|---|---|---|---|---|---|---|---|---|---|---|---|---|---|
| CLIP I2T | B | 39.9 | 22.1 | 34.5 | 18.7 | 15.7 | 8.1 | 16.1 | 9.1 | 22.4 | 12.2 | 34.9 | 21.2 | 40.0 | 26.0 | 37.3 | 23.4 |
| CLIP Fusion | B | 67.5 | 47.7 | 26.3 | 10.5 | 50.9 | 39.0 | 14.6 | 5.2 | 28.5 | 12.2 | 36.2 | 21.8 | 18.7 | 7.7 | 24.7 | 13.5 |
| CLIP2CLIP | B | 52.3 | 30.3 | 37.9 | 20.5 | 44.9 | 26.7 | 27.1 | 15.0 | **30.1** | **16.9** | 56.2 | 33.7 | 45.3 | 28.1 | **50.2** | **30.7** |
| CLIP I2T | L | 46.7 | 27.1 | 42.8 | 24.5 | 14.9 | 7.9 | 15.9 | 9.2 | 23.1 | 12.9 | 38.5 | 22.8 | 45.5 | 30.4 | 41.8 | 26.1 |
| CLIP Fusion | L | 70.1 | 51.3 | 37.8 | 16.8 | 57.8 | 43.6 | 18.1 | 7.2 | **35.5** | 16.8 | 48.6 | 31.2 | 29.0 | 13.4 | 36.4 | 18.8 |
| CLIP2CLIP | L | 54.4 | 31.8 | 43.8 | 24.2 | 40.6 | 23.3 | 31.6 | 17.6 | 33.6 | **18.8** | 56.6 | 33.4 | 51.4 | 32.5 | **53.9** | **33.0** |

Table 5: We report top methods on the OVEN-Wiki **test** set and **gold evaluation** set. Additional metrics are reported, *i.e.*, Recall@10 (R10) and Mean Reciprocal Ranks@10 (MRR10).

| Entity Repr. Image | Title | Summary | Entity Split SEEN | UNSEEN | HM | Query Split SEEN | UNSEEN | HM | Overall HM |
|---|---|---|---|---|---|---|---|---|---|
| ● | - | - | 3.99 | 3.17 | 3.53 | 2.46 | 0.96 | 1.38 | 1.99 |
| - | ● | - | 17.33 | 14.75 | 15.94 | 8.64 | 8.90 | 8.77 | 11.32 |
| - | - | ● | 18.76 | 17.25 | 17.97 | 6.75 | 8.39 | 7.48 | 10.56 |
| ● | ● | - | 18.19 | 15.42 | 16.69 | 10.35 | 9.34 | 9.82 | **12.36** |
| ● | - | ● | 19.37 | 17.47 | 18.37 | 7.88 | 8.05 | 7.96 | 11.11 |
| - | ● | ● | 18.53 | 16.86 | 17.66 | 5.96 | 8.67 | 7.06 | 10.09 |
| ● | ● | ● | 18.62 | 17.37 | 17.97 | 8.66 | 8.99 | 8.82 | 11.84 |

Table 6: Ablation study on CLIP2CLIP, with different entity representation.

**Error analysis** We show top 5 predictions of CLIP2CLIP on OVEN-Wiki in Table 7, with the left three examples being the erroneous predictions and the right example the correct prediction. From left two examples, CLIP2CLIP made mistakes among hard negative entities similar to the ground truth. On the third example, CLIP2CLIP seems to fail to reason about the image and the query jointly.

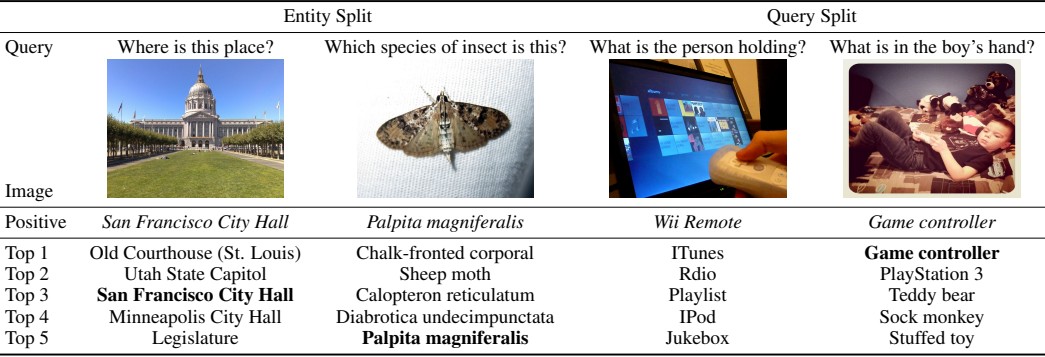

| | Entity Split | | Query Split | |
|---|---|---|---|---|
| Query | Where is this place? | Which species of insect is this? | What is the person holding? | What is in the boy's hand? |
| Image | | | | |
| Positive | *San Francisco City Hall* | *Palpita magniferalis* | *Wii Remote* | *Game controller* |
| Top 1 | Old Courthouse (St. Louis) | Chalk-fronted corporal | ITunes | **Game controller** |
| Top 2 | Utah State Capitol | Sheep moth | Rdio | PlayStation 3 |
| Top 3 | **San Francisco City Hall** | Calopteron reticulatum | Playlist | Teddy bear |
| Top 4 | Minneapolis City Hall | Diabrotica undecimpunctata | IPod | Sock monkey |
| Top 5 | Legislature | **Palpita magniferalis** | Jukebox | Stuffed toy |

Table 7: Error Analysis on the CLIP2CLIP Model.

