# OpenReview forum: "Open-domain Visual Entity Linking"
_ICLR.cc/2023/Conference — Submitted to ICLR 2023_

### Official Review · Reviewer_iTQw · 2022-10-23

**Confidence:** 4
**Clarity, Quality, Novelty And Reproducibility:** The paper is clear. The task is novel…
**Correctness:** 4
**Technical Novelty And Significance:** 2
**Empirical Novelty And Significance:** 3
**Recommendation:** 5

**Strength And Weaknesses:**

STRENGTHS
- The task is interesting without a doubt.
- The human annotated set can be a high quality evaluation set for the task.
- The baseline models are reasonable.

WEAKNESSES
- No performance upper bound. It is not clear if this problem is even solvable. In the first example in Figure 1, how can we expect the model to infer the plane manufacturer? There does not seem to be enough information in all modalities. (I think this example comes from the gold set? I wonder how the human annotators deemed it correct.)
- On a related note, the baseline performance is really weak. This needs to be addressed in conjunction with the task feasibility.
- Other than the gold set, the dataset itself is automatically constructed with basic means (Wiki API string matching + off-the-shelf text-only linker). The impact and implications of the flawed finetuning are not discussed.
- I couldn't find some obvious baselines, such as doing only image-image or text-text matching.
- The generative baseline SimplerVLM can/should do the same constrained decoding to ensure that the prediction is a valid entity.



[Post-response]

Thanks for the response. I think the human performance is a valuable addition. I still think the paper could make the feasibility and baseline components more substantial (e.g., an actual analysis of impossible examples, instead of invoking existing datasets). I'd like to keep my score.

**Summary Of The Paper:**

The paper presents a dataset for vision-language entity linking.

Both queries (q) and entities (e) have image and text component, so a score needs to be defined for the quadruple (img(q), txt(q), img(e), txt(e)).

The training(~5m queries)/validation/test splits are automatically constructed from existing image recognition/retrieval and visual QA datasets. Separately, the paper provides ~5k manually corrected gold examples for evaluation.

The main baseline is mixed-modality dual encoders based on CLIP-large, achieving the harmonic mean of seen/unseen accuracies of ~15 on dev/test and ~20 on gold.



**Summary Of The Review:**

The paper identifies the interesting problem of vision-language entity linking and provides a dataset, but the utility of the dataset except for the manually annotated portion is somewhat unclear and the paper doesn't get to the bottom of the problem (i.e., not answering natural questions like what is the best possible performance).

---

> ### Author Response · Authors · 2022-11-18
> **Response to Official Review of Paper3777 by Reviewer iTQw (2/2)**
>
> ***[4]*** _“I couldn't find some obvious baselines, such as doing only image-image or text-text matching.”_
>
> Thanks for pointing out the two baselines. We added an ablation study (see Table 1 of the revised paper) to show results on image2image matching and text2text matching, using the pre-trained CLIP model (denoted as CLIP I2I and CLIP T2T). The results have shown that text2text matching gets close to 0% accuracy and image2image matching gets 1.5% accuracy, which is significantly worse than the fine-tuned best model.
>
> ***[5]*** _”The generative baseline SimplerVLM can/should do the same constrained decoding to ensure that the prediction is a valid entity.”_
>
> Thanks for the suggestion. We agree that constrained decoding might potentially benefit the auto-regressive entity linker as it helps eliminate the invalid possibility. However, we note that it would not be able to incorporate the multi-modal entity information (e.g. infobox images, Wikipedia text) for SimplerVLM just through constrained decoding.
>
> Due to the limited rebuttal time period, we can not make the constrained decoding work properly in time, it requires a non-trivial amount of engineering efforts. Instead, we approximate it via adding a BM25 matching phase for SimplerVLM, which ensures SimplerVLM’s prediction is grounded to valid 105K Wikipedia entities. As a result, it improves the zero-shot SimplerVLM from 0.3% to 0.8% (and does not improve the fine-tuned SimplerVLM). This indicates the possibility that fixing decoding method for a weak pretraining auto-regressive model is not sufficient.

---

> ### Author Response · Authors · 2022-11-18
> **Response to Official Review of Paper3777 by Reviewer iTQw (1/2)**
>
> We would like to thank you for your constructive feedbacks, and below are our responses to your questions:
>
> ***[1]*** _“No performance upper bound. It is not clear if this problem is even solvable. In the first example in Figure 1, how can we expect the model to infer the plane manufacturer? There does not seem to be enough information in all modalities. (I think this example comes from the gold set? I wonder how the human annotators deemed it correct.)”._
>
> ***[Is the task solvable?]*** In OVEN, all current examples are derived from existing labeled datasets, hence the feasibility of recognizing the classes from visual features have been validated before in prior work. For example, the aircraft manufacturer example is derived from a widely used image recognition dataset called FGVC-Aircraft. The key reason that models are getting lower numbers  (in accuracy) is because OVEN additionally requires the models to understand the query text, to search in a large entity space, and to generalize to novel & unseen entities, which is fundamentally more difficult than its original form.
>
> ***[human upper bound]*** To provide supporting evidence for the task feasibility, we perform a human performance study on a subset of the gold evaluation set (a total of 100 examples), with a total of 3 annotators (each data is annotated by 2 annotators). This task is very challenging for humans, so we allow the annotators to use all existing online tools available. In particular, we ask the annotators to read the image and the query, and use tools like Google Search, Wikipedia Search and Image Search to perform detailed research on each example. The annotators are also ensured to select only the Wikipedia entities among 105K candidates. Even with the help of all available tools, it takes on average 254 seconds for each annotator to finish one annotation. The averaged human performance is 76% (measured in average accuracy), whereas the best system achieves 25% on the same subset. Note that human study is not necessarily the model upper-bound, but it shows that there is sufficient headroom available for this task.
>
> ***[2]*** _“On a related note, the baseline performance is really weak. This needs to be addressed in conjunction with the task feasibility.”_
>
> We would like to point out that the low performance of the various models is mainly due to the challenges brought by OVEN. There are few classification tasks that require considering 105k possible output labels, and the requirement to generalize unseen entities is very challenging as well. Moreover, OVEN requires model to jointly encoder *both* images and text on queries *and* documents. To the best of our knowledge, there is no such a pre-trained multi-modal model that has this capability at the time of this paper submission. However, we agree that there is significant headroom for the task. As aforementioned human performance study,  with the help of search engines and other tools, humans can reach 76% accuracy, which is much higher than the performance of the current systems (25% accuracy).
>
> ***[3]*** _“Other than the gold set, the dataset itself is automatically constructed with basic means (Wiki API string matching + off-the-shelf text-only linker). The impact and implications of the flawed fine-tuning are not discussed.”_
>
> Note that our training data is not purely constructed automatically. One critical step in our annotator is to use rules to filter out examples that are possibly incorrect due to language ambiguity. This is a complicated and an iterative procedure. As a result, our training data in fact has pretty high precision. When validated by annotators, annotators found that between only 8 to 12% of examples require entity link correction. Indeed,  highlight that fine-tuning on our automatically generated dataset is improving the model's performance on the gold evaluation dataset. Specifically, the zero-shot CLIP-based method gets 13.1%, and the fine-tuned CLIP-based model improves to 17.5%.
>
> Ultimately, we believe that the central capabilities of OVEN should be learned from the large-scale pre-training, where new models and algorithms need to be designed to learn correspondence between multimodal queries, and multi-modal knowledge representation (such as  Wikipedia, general Web pages, and documents). In this manner, the role of fine-tuning data is mainly to provide in-domain knowledge for fast adaptation of pre-trained foundation models.

---

### Official Review · Reviewer_hwGY · 2022-10-24

**Confidence:** 5
**Correctness:** 2
**Technical Novelty And Significance:** 2
**Empirical Novelty And Significance:** 2
**Recommendation:** 3

**Clarity, Quality, Novelty And Reproducibility:**

The paper is mostly clear and easy to read. Though there are a few language errors that need to be fixed such as:
Given these inputs the goal -> Given these inputs, the goal
For final performance, -> For the final performance,
De Cao et al. (2021) has shown -> De Cao et al. (2021) have shown
large scale contrastive pretraining -> large-scale contrastive pretraining

The proposed task, dataset, and model are not completely novel, as discussed in the weakness section. Regarding reproducibility, sufficient details are there in the paper to implement the models. Do the paper also plan to make dataset and baseline implementations public?

**Strength And Weaknesses:**

Strengths:

+ A large-scale visual entity linking benchmark namely OVEN-Wiki has been presented. Visual entity linking (especially at large-scale) is an underexplored problem in the literature. It has utility in many downstream tasks such as knowledge-aware VQA, knowledge-aware image captioning, and in general better scene interpretation. Further, visual entity linking is also important in many applications such as news search and e-commerce. Therefore, this paper has definitely some value.

+ Baselines using state-of-the-art pre-trained models have been presented. Evaluations are done for both seen and unseen entities.

Weakness:

- Literature and designed baselines are weaker. The following recent work also presents a dataset and model for visual entity linking task:

[A] Qiushuo Zheng, Hao Wen, Meng Wang, Guilin Qi: Visual Entity Linking via Multi-modal Learning. Data Intell. 4(1): 1-19 (2022)

Their model looks closer to state-of-the-art entity-linking models in NLP literature. Why such models are not considered as one of the competitive baselines? Even the dataset presented in this paper can also be used to benchmark methods. Further, arguments in the paper such as “opens up possibilities to answer questions on entities that have not been learned before, and are thus UNSEEN.” Is not completely correct, as the following paper performs zero-shot VQA:

[B] Zhuo Chen, Jiaoyan Chen, Yuxia Geng, Jeff Z. Pan, Zonggang Yuan, Huajun Chen: Zero-Shot Visual Question Answering Using Knowledge Graph. ISWC 2021: 146-162

Further, in a recent work namely WebQA, VQA on unseen object categories has also been shown. They also have proposed a dataset that has large coverage of visual entities. (One difference I see is they heavily rely on captions as well rather than completely doing visual reasoning). Nevertheless, it is a very relevant paper:

[C] Yingshan Chang, Guihong Cao, Mridu Narang, Jianfeng Gao, Hisami Suzuki, Yonatan Bisk:
WebQA: Multihop and Multimodal QA. CVPR 2022: 16474-16483

- Regarding the proposed dataset: The dataset surely has some merits, especially the scale and diversity. However, it is not clear why one of the most prominent visual entities namely PER (or public figures) is not considered. I understand that including public figures also make the problem closer to face recognition at a very large scale. But, possibly it might have yielded a stronger dataset. There are several such public image datasets such as KVQA, oxford’s people in places that could have been used. Further, In NLP literature, the context has played important role in linking entities. The same may also be very important in visual entity linking. I am not sure if such things can be explored in the proposed dataset. Furthermore, how much role does natural language understanding play in the task? Are there complex queries ( as the paper says ambiguous queries are rewritten)? Isn’t it good to separate out visual entity linking from NLU tasks?

- Regarding evaluation: Traditionally, rank@K and mean reciprocal ranks are used to measure entity linking. In problems like visual entity linking it is often important to get top-k predictions correct. It is not clarified why f-scores are preferred over such established and seemingly more suitable evaluation measures.

- Does the paper consider some hierarchical approach where first-level high-level categories (such as animal, plant, building, etc.) are classified? I believe such an approach may improve performance.


**Summary Of The Paper:**

The paper presents open-domain visual entity linking and contributes a large-scale dataset containing 4.89 Million and 730K examples in train and test sets respectively. The paper also presents several baselines for the task using state-of-the-art approaches. Visual entity linking for both seen and unseen entities have been reported.

**Summary Of The Review:**

The presented task is important and underexplored. The proposed dataset is large-scale but its coverage is limited and does not really try to solve visual entity linking as a standalone task. Evaluation measures are not well justified and some important baselines and literature have been dropped.

Post-author response:
I am not fully satisfied with the difference between VELD [A] and the proposed work provided in the author's response. Though the images in VELD come with captions and the proposed model (full) uses them, the obvious ablation for [A] is not to use any textual feature. They indeed perform such ablation, for example, in Table 3 in [A]. Therefore, VELD seems very relevant both model and dataset-wise. The setting proposed in this work is evaluating VQA and only implicitly visual entity linking. It would be great to perform VEL without any textual input (neither caption nor question, just an image and KG). With such a setting and appropriately defined task, the value of the proposed dataset will significantly increase.

It is good to see the response has now reported R@K and MRR-based results.

I still have concerns about the baseline experiments. They are rather weak and do not really represent SOTA entity linking literature or multimodal entity linking literature. A simple hierarchical approach might have been a good competitor. For example, if a person is what is required to identify, I would rather prefer to use a specialized model for face recognition, and the same applies to other categories.

With these concerns, I am inclined toward sticking to my original rating.

---

> ### Author Response · Authors · 2022-11-18
> **Response to Official Review of Paper3777 by Reviewer hwGY (2/2)**
>
> ***[3]*** _“Regarding the proposed dataset: The dataset surely has some merits, especially the scale and diversity. However, it is not clear why one of the most prominent visual entities namely PER (or public figures) is not considered.”_
>
> We do not particularly ignore any person entities. In the set of entities with labeled examples, there are about 300 - 400 entities under the supercategory of “person”. Among the total 105K entity candidates in KB, there are more than 5k person entities. Moreover, we believe that OVEN is only taking an initial small step and we hope more community work would follow up to produce the larger scale, and more diverse datasets for studying visual entity linking.
>
> ***[4]*** _“Further, In NLP literature, the context has played an important role in linking entities. The same may also be very important in visual entity linking. I am not sure if such things can be explored in the proposed dataset. Furthermore, how much role does natural language understanding play in the task? Are there complex queries ( as the paper says ambiguous queries are rewritten)? Isn’t it good to separate out visual entity linking from NLU tasks?”_
>
> As aforementioned, the text query in OVEN is a question that asks which part of the image to query, and does not mention the name of the entity at all, so there is not a lot of meaningful language context that can be leveraged. In fact, our studied text-only entity linker gets zero accuracy on the OVEN-wiki dataset.
>
> ***[5]*** _“Regarding evaluation: why not rank@K and mean reciprocal ranks?”_
>
> We thank the suggestion on alternative evaluation metrics. To make our evaluation more comprehensive, we have revised our paper to include both recall@10 and MRR@10 as additional evaluation metrics in the Appendix (Table 5 and Table 6). The overall trend is similar to the results in the main table.
>
> ***[6]*** _”Does the paper consider some hierarchical approach where first-level high-level categories (such as animal, plant, building, etc.) are classified? I believe such an approach may improve performance.”_
>
> This is a good idea, we hope that with OVEN the research community can explore this opportunity in the future.
>
> ***[7]*** _“Do the paper also plan to make dataset and baseline implementations public?”_
>
> Yes, we are working on releasing the dataset and baseline methods in this paper to public

---

> ### Author Response · Authors · 2022-11-18
> **Response to Official Review of Paper3777 by Reviewer hwGY (1/2)**
>
> We would like to thank you for the detailed review and valuable comments. However, we believe that there could have been a mis-understanding of the proposed OVEN task, and the OVEN-wiki dataset.
>
> Below are our concrete responses to your questions:
>
> ***[1]*** _“The following recent work also presents a dataset and model for visual entity linking tasks: [A] … Their model looks closer to state-of-the-art entity-linking models in NLP literature. Why are such models not considered as one of the competitive baselines? Even the dataset presented in this paper can also be used to benchmark methods.”_
>
> OVEN is significantly different from the mentioned paper [A] because the queries in OVEN do not explicitly mention the name of the entity at all. To facilitate a better understanding, here is a demonstrative example that illustrate the differences among (1) text entity linking, (2) entity linking in [A] VELD, and (3) OVEN:
>
> | | Standard Text Entity Linking | Entity linking in [A] VELD dataset | OVEN |
> |----------------|---------------------|------------|-------------------|
> | Input Image | None | [Image Link](https://en.wikipedia.org/wiki/Welsh_Corgi#/media/File:Kelsey_n_Penny_(Welsh_Corgis).jpg) | [Image Link](https://en.wikipedia.org/wiki/Welsh_Corgi#/media/File:Kelsey_n_Penny_(Welsh_Corgis).jpg) |
> | Input Text | “A Cardigan Welsh Corgi (left) and a Pembroke Welsh Corgi (right)” | “A Cardigan Welsh Corgi (left) and a Pembroke Welsh Corgi (right)” | "What is the breed of the dog on the left?”|
>
> As we can see, the input text to (1) or (2) is caption text that contains sufficient entity information to make an entity linking prediction, whereas the query for (3) does not reveal such information. Instead, OVEN requires strong visual understanding and reasoning ability to perform a successful inference. Therefore, applying state-of-the-art entity-linking models (i.e, GENRE) in NLP literature focusing only on text get  0.1% accuracy in OVEN (human annotated gold evaluation set). For a comparison, such text-only model (i.e. GENRE) can get 73.3% top-1 accuracy on the [A] VELD dataset.
>
> ***[2]*** _“Further, arguments in the paper such as “opens up possibilities to answer questions on entities that have not been learned before, and are thus UNSEEN.” Is not completely correct, as the following paper performs zero-shot VQA… Further, in a recent work namely WebQA, VQA on unseen object categories has also been shown. They also have proposed a dataset that has large coverage of visual entities. (One difference I see is they heavily rely on captions as well rather than completely doing visual reasoning).”_
>
> Thanks for comments and pointing out the related literature. The claim of “open up …” is not the main argument of our paper and we have revised our text to be more accurate. We also included a discussion in the related work for the mentioned [B] zero-shot VQA and [C] WebQA datasets.
>
> We would like to stress that while OVEN is related to VQA, OVEN is not a VQA task. VQA models output free-form answers, whereas OVEN models are required to output the entity name inside a KB (same as text entity linking tasks). Moreover, as pointed out by the reviewer, OVEN encourages visual reasoning on a wide range of different entities uniformly, where existing VQA tasks often have a more limited coverage of the entities.
>
> Finally, while the issue of unseen classes has been covered by prior VQA and classifications dataset, the amount and the scale of unseen entities in OVEN  (50\% labels are unseen entities, more than 400K examples) is quite large. Therefore, OVEN enforces the models to put generalization as their first priority.

---

### Official Review · Reviewer_8tsg · 2022-10-24

**Confidence:** 5
**Correctness:** 3
**Technical Novelty And Significance:** 3
**Empirical Novelty And Significance:** 3
**Recommendation:** 6

**Clarity, Quality, Novelty And Reproducibility:**

The paper clarified its contributions and approach most clearly.
In general, the paper is novel, can be reproduced and is of fair quality.

**Strength And Weaknesses:**

*[Strength]*
1. This work constructs a new knowledge-based dataset, reflecting a more realistic scenario that there will never be enough training data to cover all knowledge-base entities, especially when the number of knowledge-base entities is constantly growing.
2. It defines a new evaluation metric for the proposed task.

*[Weakness]*
1. In equation 1, it will be better if add the concept that the prediction may be unseen in the question and image input.
2. What are the two questions in Figure 2?
3. Is the example mentioned in the paper defined as image-text pairs? If so, does that mean the unseen examples partially include unseen entities?
4. Also, in the experiments section, there is seen/unseen entity split and query split, what is the relation between them and the seen/unseen examples in Figure 3?

**Summary Of The Paper:**

This paper mainly focuses on linking to the entity out of all entities in the knowledge base. A benchmark dataset is collected by linking all existing labels to Wikipedia entities when possible, using a state-of-the-art entity linking system and human annotators, creating a diverse and unified label space. It requires models to recognize and link visual content to both a small set of seen entities as well as a much larger set of unseen entities. Also, it requires models to generalize to previously unseen intents that may require more fine-grained reasoning.

**Summary Of The Review:**

This paper formally defined a new task that reflects a more realistic scenario. And it collected a new dataset and proposed corresponding novel evaluation metrics.
Even though there are some unclear clarifications, in general, it benefits the community.
Overall, I'm leaning to accept it.

---

> ### Author Response · Authors · 2022-11-18
> **Response to Official Review of Paper3777 by Reviewer 8tsg**
>
> We would like to thank you for the valuable and encouraging review, and below are our responses to your questions:
>
> ***[1]*** _”In equation 1, it will be better if you add the concept that the prediction may be unseen in the question and image input.”_
>
> We have adjusted our notation to accommodate the requested change, and denoted the entity set \mathcal{E} as the combination of \mathcal{E}_{\seen} and \mathcal{E}_{\unseen}. Please see the text around equation 1 for the updates.
>
>
> ***[2]*** _”What are the two questions in Figure 2?”_
> The text query is “What is the model of the vehicle?” and “Who manufactured the airplane?” for the left and right examples, respectively.
>
> In Figure 2, our goal is to illustrate that OVEN is challenging to a recognition model such as CLIP (which performs image-to-text matching), because of the two reasons below:
> - (1) [Left figure] With an increasing number of candidate entities, CLIP faces a much more difficult task than recognition within a pre-defined small label space.
> - (2) [Right figure] Without access to the query, models such as CLIP can not choose one entity out of all candidates as many candidates can be appropriate to the image. Only contextualization with the query can eliminate the ambiguity in recognition.
>
> ***[3]*** _”Is the example mentioned in the paper defined as image-text pairs? If so, does that mean the unseen examples partially include unseen entities?”_
>
> We are not 100% sure about the meaning of this question. Our educated guess is that the reviewer would like us to clarify the role of the text query in OVEN. In OVEN, the text is only a query (not a caption), and does not contain any named entities. If we only use text (without the image) to perform entity linking using state-of-the-art entity linker (i.e., GENRE) , it would lead to  0.1% accuracy (on gold evaluation set). Therefore, we believe that the text does not provide sufficient information to reason about the unseen entities.
>
> ***[4]*** _”Also, in the experiments section, there is seen/unseen entity split and query split, what is the relation between them and the seen/unseen examples in Figure 3?”_
>
> For train, validation, and test data, we have two non-overlapping data splits, i.e., the entity splits (derived from recognition datasets), and the query splits (derived from Visual QA datasets). Both the entity and query splits have examples covering SEEN and UNSEEN “visual entities”.
>
> The key difference is that data in the entity split uses templated query text, and covers more diverse and fine-tuned visual entities (e.g. car models, animal species, landmarks); whereas data in the query split uses human generated natural language queries (obtained from VQA datasets), but often covers more frequent and less diverse visual entities (e.g. common objects).

---

### Official Review · Reviewer_cfNC · 2022-10-26

**Confidence:** 4
**Correctness:** 4
**Technical Novelty And Significance:** 3
**Empirical Novelty And Significance:** 3
**Recommendation:** 8

**Clarity, Quality, Novelty And Reproducibility:**

The paper is clear and of high quality. The authors provide numerous results in their supplementary material.
The dataset is highly novel and will be a useful multimodal reasoning task with important downstream use cases.
The paper describes the baselines and method in sufficient detail to reproduce the approach the authors have taken. The dataset will  be publicly released and thus the paper is reproducible.

**Strength And Weaknesses:**

[Strengths]
The proposed task (OVEN) is highly interesting, realistic, and likely to be of high-impact. For example, methods developed on OVEN could reasonably be used to aid in many multimodal reasoning task, such as question answering or any tasks requiring reasoning over external knowledge. The proposed task is also highly challenging for most existing methods. For example, the Wikipedia articles often feature text and multiple images. A successful model should integrate the knowledge from the multiple images on the Wikipedia page to ensure a successful link.

The authors demonstrate that OVEN is an extremely challenging task requiring complex reasoning to adequately perform well. OVEN requires determining which of ~100k Wikipedia entities to link to. This is an enormous possible label space - scary even to those who work in zero-shot learning. Thus, the problem is a very interesting avenue for future research and is likely to get much interest.

The dataset is cleverly constructed and well-designed. The authors clearly invested significant resources in its construction, employing ~30 human linkers to gather thousands of ground truth links. The authors also cleverly re-use existing datasets from VQA and object recognition and note that some datasets feature diverse visual entities, while others feature diverse language queries - thus the dataset features an excellent balance of diversity.

The authors propose a number of baseline methods on their dataset that make use of CLIP or SimVLM and perform a detailed experimental analysis and ablation study. The authors make a number of key findings. For example, that not-all fine-tuned models were better than their untuned versions on the task. The authors' ablations are important and explore important aspects of the task or model such as how important input representations are for OVEN, the impact of fine-tuning on performance, and provide error analysis.

[Weaknesses]
One weakness is the baseline models are quite simple / weak. I recognize that OVEN is primarily a dataset paper, however and that the primary contribution does not come from the baseline model, but rather the effort expended creating the dataset and formalizing the task.

Perhaps I missed it, but how accurate are the human annotators at this task? Have the authors computed an inter-annotator agreement score? How often do humans agree on the Wikipedia entity link for the same image and query?

**Summary Of The Paper:**

In this paper, the authors propose a new task, similar to the traditional text entity linking text where a textual entity is linked to its associated entry in Wikipedia. In this work, the authors propose to link "visual" entities to Wikipedia. Because the same entity could reasonably be linked to different Wikipedia entries (a photo of a car could be linked to "car" or the particular model), the authors propose to unambiguously link to Wikipedia by also incorporating a text query (e.g. "what model is it?").
To make progress on this task, the authors propose a new dataset, called OVEN-Wiki. The proposed dataset is constructed from existing datasets wherein all of the labels within the dataset are grounded to Wikipedia. In addition, the authors employed over 30 human annotators to annotate a subset of OVEN-Wiki.
To create a benchmark on this task, the authors formulate and evaluate a number of baselines using exsting VL models. They experiment and analyze the performance of using models like CLIP zero-shot at this task vs. finetuning models on their dataset.
The most successful baseline proposed by the authors leverages image-text pairs from the Wikipedia article and query image-text pair and essentially takes cosine distance to find the best match.

**Summary Of The Review:**

The authors propose an interesting, impactful, and challenging new task called OVEN. They contribute a large-scale benchmark for this task and provide initial baseline results. The contributed dataset is of high quality and is the result of a large-scale human annotation effort. The baseline models are less novel, but are sufficient to establish a "baseline" for future work to compare against and beat. In sum, the paper makes a significant contribution likely to be of use to a wider audience and should be accepted.

---

> ### Author Response · Authors · 2022-11-18
> **Response to Official Review of Paper3777 by Reviewer cfNC**
>
> We would like to thank you for the valuable and encouraging review, and below are our responses to your questions:
>
> ***[1]*** _“Simple/Weak Baselines”_
>
> We agree that there is significant headroom for the task, and we would like to point out that the low performance of the various models is also due to the challenges brought by OVEN. There are very few classification tasks that require considering over 105k possible output labels, and the requirement to generalize to unseen entities is challenging as well. Moreover, OVEN requires the model to jointly encode *both* images and text on the query side (context image+text query) *and* the Wikipedia entity side (infobox image+Wikipedia text). To the best of our knowledge, there is no such a pre-trained multi-modal model that has this capability at the time of this paper submission.
>
> With the fact that no appropriately pretrained model exists for OVEN, CLIP-Large is close to the best public image-to-text matching model that we can find. Therefore, we mainly focus on adapting CLIP for OVEN. While other pretrained multimodal encoders (e.g. UNITER) can be applied for retrieval as well, we found out that fine-tuning such encoders as dual encoders could  not approach good results on unseen entities, as shown by the Dual-SimplerVLM baseline.
>
> Among our baselines, the enc-dec model (SimplerVLM) is the weakest, because the decoder can decode strings that do not match any entity name in the KB. To make sure the enc-dec model always outputs an entity in our KB, we add a post processing step to ground the decoded string to entities in the KB, using BM25 as the similarity metric (the decoded string is matched to its nearest text among the 105K valid entity names). As a result, the accuracy improves from 0.3% to 0.8%, indicating the possibility that fixing decoding method for a weak pretraining auto-regressive model is not sufficient.
>
> ***[2]*** _“Perhaps I missed it, but how accurate are the human annotators at this task? Have the authors computed an inter-annotator agreement score? How often do humans agree on the Wikipedia entity link for the same image and query?”_
>
> During annotation, we ask the annotations to verify the entities produced by our procedure. Out of all annotations, 88% of the examples all three annotators agree on the annotations. We *only* kept the ones that all three annotators agree in our gold evaluation set.
>
> In addition, we also perform a human performance study on a subset of the gold evaluation set (a total of 100 examples), with a total of 3 annotators (each data is annotated by 2 annotators). This task is very challenging for humans, so we allow the annotators to use all existing online tools available. In particular, we ask the annotators to read the image and the query, and use tools like Google Search, Wikipedia Search and Image Search to perform detailed research on each example. The annotators are also ensured to select only the Wikipedia entities among 105K candidates. Even with the help of all available tools, it takes on average 254 seconds for each annotator to finish one annotation. The averaged human performance is 76% (measured in average accuracy), whereas the best system achieves 25% on the same subset.

---

### Author Response · Authors · 2022-11-18
**General Response to All Reviewers and ACs**

We would like to thank all reviewers for their constructive feedback. We are pleased to find that reviewers think our paper is “highly interesting, realistic, and likely to be of high-impact” (Reviewer cfNC), “interesting without a doubt” (Reviewer iTQw), “more realistic scenario with novel evaluation” (Reviewer 8tsg), and recognize our constructed dataset to be “large-scale, diverse” as well as to address an “underexplored problem in the literature” (Reviewer hwGY).

For the comments each reviewer has, we have prepared responses to address them accordingly. Among all of them, we found an important mis-understanding which confused OVEN with other entities linking (EL) literature. To clarify this, we provide an illustrative example as below:

| | Standard Text Entity Linking | Entity linking in [A] VELD dataset | OVEN |
|----------------|---------------------|------------|-------------------|
| Input Image | None | [Image Link](https://en.wikipedia.org/wiki/Welsh_Corgi#/media/File:Kelsey_n_Penny_(Welsh_Corgis).jpg) | [Image Link](https://en.wikipedia.org/wiki/Welsh_Corgi#/media/File:Kelsey_n_Penny_(Welsh_Corgis).jpg) |
| Input Text | “A Cardigan Welsh Corgi (left) and a Pembroke Welsh Corgi (right)” | “A Cardigan Welsh Corgi (left) and a Pembroke Welsh Corgi (right)” | "What is the breed of the dog on the left?”|

_Note: [A] Qiushuo Zheng, Hao Wen, Meng Wang, Guilin Qi: Visual Entity Linking via Multi-modal Learning. Data Intell. 4(1): 1-19 (2022)_

OVEN is mainly different from other EL tasks as the text query of OVEN does not explicitly contain sufficient information about the entity’s name. To supplement this, we evaluated GENRE (SoTA text entity linking model) on the text query of OVEN (on human annotated gold evaluation set), which archives near zero performance (0.1%). This suggests that visual understanding and reasoning is most critical in solving the task of OVEN.

As suggested by the reviewers, we have also made the following changes to our paper:
1. As suggested by the reviewer iTQw, we included an additional human study in section 5.1 to show the task feasibility of OVEN.
2. As suggested by reviewer hwGY, we revised section 2.1 to adjust our claim.
3. As suggested by reviewer hwGY, we included additional evaluation metrics to Table 4 and Table 5 in the Appendix.
4. As suggested by reviewer hwGY, we extend section 6 to add more discussions on related works. We moved the literature review on entity linking (originally in the Appendix) to section 6 to compare OVEN to both traditional entity linking and visually assisted entity linking.
5. Due to space limitation, we moved the error analysis and qualitative examples from the main text to the Appendix.
6. We fixed minor typos and re-formatted the tables and figures to improve the clarity.

---

### Decision · Program_Chairs · 2023-01-20

**Decision:**

Reject

**Justification For Why Not Higher Score:**

All the reviewers and AC appreciated the contributions of the benchmark, however, it is not comprehensive enough, e.g., lacking strong baselines, task potentials, and challenges. Therefore, AC believes that the readers in the community cannot truly benefit from the work rather than just the acknowledgment of the benchmark.

**Justification For Why Not Lower Score:**

N/A

**Metareview: Summary, Strengths And Weaknesses:**

The paper proposes a benchmark for linking Wiki knowledge to a visual entity (e.g., an image with a label) given a query specifying what domain knowledge the entity is linking to. The authors provide human annotation ground-truth and several baselines for the proposed benchmark.

Strength:
A new benchmark called OVEN-Wiki has been proposed for visual entity linking that may support knowledge based vision-language tasks such as KVQA and Knowledge image captioning.

Weakness:
As a dataset paper, it lacks a significant body of evaluations, metrics, potential challenges, and risks. The weakness has been raised by all the reviewers.

**Summary Of Ac-Reviewer Meeting:**

The merits of the contribution of the new benchmark are appreciated by all the reviewers. Reviewers with positive ratings consider that the contribution of the benchmark itself is more important than the technical contributions of the papers, and those with negative ratings clearly outweigh the latter more than the former. AC read the paper and the rebuttal and feels that even for the standard of a benchmark paper, this paper is far from comprehensive, for example, lacking strong baselines, task potentials, and challenges.